Monitoring body condition score of reintroduced banteng (Bos javanicus D’Alton, 1923) into Salakphra Wildlife Sanctuary, Thailand

Kongsurakan Praeploy 1
http://orcid.org/0000-0002-1984-5236 Chaiyarat Rattanawat 1 rattanawat.cha@mahidol.ac.th
Nakbun Seree 2
Thongthip Nikorn 3 4 5
Anuracpreeda Panat 6
1 Wildlife and Plants Research Center, Faculty of Environment and Resource Studies, Mahidol University , Phuttamonthon, Nakhon Pathom , Thailand
2 Khao Nampu Nature and Wildlife Education Center, Department of National Parks, Wildlife and Plant Conservation , Kanchanaburi , Thailand
3 Faculty of Veterinary Medicine, Kasetsart University , Kamphaeng Saen Campus, Nakhon Pathom , Thailand
4 Center for Agricultural Biotechnology, Kasetsart University , Kamphaeng Saen Campus, Nakhon Pathom , Thailand
5 Center of Excellence on Agricultural Biotechnology, Science and Technology Postgraduate Education and Research Department Commission on Higher Education, Ministry of Education (AG-BIO/PERDO-CHE) , Bangkok , Thailand
6 Institute of Molecular Biosciences, Mahidol University , Phuttamonthon, Nakhon Pathom , Thailand
Pimm Stuart
Electronic publication date: 2020 Apr 23
Publication date: 2020
Volume: 8
Electronic Location ID: e9041
Received 2019 Sep 25; Accepted 2020 Apr 1
Copyright: © 2020 Kongsurakan et al.
Copyright year: 2020
Copyright holder: Kongsurakan et al.
License: This is an open access article distributed under the terms of the Creative Commons Attribution License, which permits unrestricted use, distribution, reproduction and adaptation in any medium and for any purpose provided that it is properly attributed. For attribution, the original author(s), title, publication source (PeerJ) and either DOI or URL of the article must be cited.
License URL: https://creativecommons.org/licenses/by/4.0/

Keywords: Banteng, Camera trap, Reintroduction, Salakphra Wildlife Sanctuary, Body condition score (BCS)

Funding: Red Bull Beverage Co., Ltd, Thailand Funding support was received from the Red Bull Beverage Co., Ltd, Thailand. There was no additional external funding received for this study. The funders had no role in study design, data collection and analysis, decision to publish, or preparation of the manuscript.

==============================
Background

Banteng (Bos javanicus d’Alton 1823) are an endangered species, highly sensitive to habitat structure and quality. In many areas, banteng were extinct and needed to be reintroduced to restore their population. Thus, understanding the responses of body condition of reintroduced banteng to their habitat was important for ensuring the sustainability of a reintroduction program. The aim of the present study was to evaluate the body condition of banteng after reintroduction into the Salakphra Wildlife Sanctuary in Thailand based on photographs from camera-traps carried out between July 2016 and November 2018.

Methods

Seven banteng were bred at the Khao Nampu Nature and Wildlife Education Center and systematically reintroduced into the Salakphra Wildlife Sanctuary in December 2015 (four) and July 2016 (three). The seven reintroduced adults and two newborns (from the 2015 group) were captured via camera traps in 2018. The body condition scoring (BCS) obtained from these photographs was used to identify the individual performance of all seven adults after their reintroduction.

Results

The BCS scores in reintroduced adult banteng, both males and females, (between 5 and 7 years old) increased significantly over time after reintroduction into a natural habitat (p < 0.05), although the BCS scores in females were not significantly different between the second and third years (p > 0.05).

Conclusions

The results from the present study suggest that camera traps are a practical tool to assess the BCS of reintroduced banteng, and can be used to monitor their condition post-release. These techniques may be appropriate for translocation programs elsewhere.

Introduction

Reintroduction is a restoration program in which animals are translocated to areas within their historic range when the population and habitat of the animal there have decreased (Conant, 1988). The role of captive breeding and reintroduction programs that aim to augment or reestablish wildlife populations has increased dramatically (Ebenhard, 1995), including the endangered banteng (Bos javanicus d’ Alton, 1823), into parts of its former natural range (Armstrong & Seddon, 2008; Moorhouse, Gelling & Macdonald, 2009; Massaro et al., 2018).

Banteng of family Bovidae are distributed in Myanmar, Laos, Vietnam, Cambodia, Borneo (the Malaysian state of Sabah and Indonesian Kalimantan), Java, Bali and Thailand (Corbett & Hill, 1992). Their lifespan is 11 years in the wild and may reach up to 20–25 years in captivity. The basic social group appears to be composed of female–juvenile units (as in other large Asian Bovini) with larger groups tending to be more-or-less temporary assemblages. Maternal herds containing several adult cows, juveniles and calves occur; these groups can often contain one or more subadult and adult males. Groups of cows without calves are also seen. For much of the year, adult banteng are largely sexually segregated and all-male groups are frequently encountered. Solitary animals tend to be mature bulls or sometimes old cows. The composition of small groups of cows with calves or juveniles and the solitary state of old individuals, may remain the same for months or even years. The composition of other small groups, particularly the unisex groups, usually varies from day to day. During the mating season the male groups disband and dominant males compete for access to receptive females (Gardner et al., 2016). The weight of banteng may reach between 600 to 800 kg in males and 590–670 kg in females. The global population of wild banteng is estimated at between 5,000 and 8,000 (Pudyatmoko, 2004). Banteng prefer more open dry deciduous forests and avoid evergreen rainforests but occupy secondary forest formations that are a result of logging and fires and enter tracts of sub-humid forest on occasions in the more humid areas of Java and Borneo (Wharton, 1968). However, the predominant habitat type is the tropical lowland dipterocarp forest in Sabah (Gardner et al., 2014).

The IUCN has listed banteng as globally endangered (Gardner et al., 2016), but the Convention on International Trade in Endangered Species of Wild Fauna and Flora (CITES) does not list banteng in its appendix (Gardner et al., 2016) since banteng that are traded in the market are genetically pure and domesticated on farms and classified as a domestic animal. Thus, the trading of this meat is not against CITES law. However, they are protected under the Wildlife Preservation and Protection Act B.E. 2562 (2019) of Thailand (Royal Thai Government Gazette, 2019). Habitat loss and degradation (Srikosamatara, 1993; Prayurasiddhi, 1997) and human disturbances (Gardner, 2014; Gardner et al., 2016; Chaiyarat et al., 2018) have significantly affected banteng and reduced their population, as has commercial hunting (Srikosamatara & Suteethorn, 1995; Chaiyarat et al., 2018) and disease transmitted by domestic cattle (B. taurus and B. indicus) that still occur in some protected areas (Chaiyarat & Srikosamatara, 2009). Accidental hybridization in the wild and active inbreeding between banteng and domestic cattle in captivity to develop the livestock industry reduces the purity of the genetic status of both the wild and captive populations (Purwantara et al., 2012).

The wild banteng population in Thailand was decreasing and estimated at only 470 in the 1990s (Srikosamatara, 1993; Srikosamatara & Suteethorn, 1995). In the Salakphra Wildlife Sanctuary, banteng were locally extinct. An example of a successful increase in a banteng population was in the Khao Khiao—Khao Chomphu Wildlife Sanctuary, Chonburi province, Thailand. However, this was an accidental introduction without systematic planning (Chaiyarat et al., 2018). Using lessons learned from this incident, the first systematic introduction of captive-bred banteng into the natural habitat was set up at the Salakphra Wildlife Sanctuary where banteng successfully adapted to the new environment (Chaiyarat et al., 2019). In 2015, the first group (two males and two females) was reintroduced in December (dry season), while the second group (two males and one female) was reintroduced in July 2016 (wet season) in the same area (Chaiyarat et al., 2019). The behavior and physiology of banteng can be altered after reintroduction into a new environment, especially by the change of diet and the need to forage. It is important to understand the health status of the population by using a body condition scoring (BCS) system that measures the health of the animals and reflects the status of the program.

Body condition scoring systems have been developed and used on many wildlife species to investigate the population integrity when their resources are limited or environment is changing (DelGiudice et al., 2011; Lane et al., 2014; Carpio et al., 2015; McWilliams & Wilson, 2015). The BCS evaluates the outer appearance of an individual’s fitness, including survival and reproduction potential, based on visual cues. The advantages of using the BCS are feasibility, simplicity and low costs (Schiffmann et al., 2017). Thus, measuring animal BCS is important and has been applied to large mammalian herbivores such as African buffalo (Syncerus caffer caffer; Ezenwa, Jolles & O’Brien, 2009), the greater one-horned rhinoceros (Rhinoceros unicorni; Heidegger et al., 2016), Asian elephants (Elephas maximus; Pokharel, Seshagiri & Sukumar, 2017), and Bornean banteng (Bos javanicus lovii; Prosser et al., 2016).

The present study aims to monitor the BCS of these captive-bred banteng in both males and females over 3 years after reintroduction into a natural habitat by using photographs from camera-traps. The BCS measurements are a simple and reliable method to score body condition in reintroduced banteng that can be performed by a standardized assessment of photographic views. The BCS can be used to monitor possible animal welfare factors such as body weight development, fertility and mortality, to improve the overall success rate of reintroductions to promote the conservation of this endangered bovid.

Materials and Methods

Study area

Salakphra Wildlife Sanctuary (14°8′37.09″N, 99°20′33.51″E, area: ~860 km2) is located in Mueang, Bo Phloi, Si Sawat and Nong Prue district, Kanchanaburi province, Thailand (Fig. 1). ArcView V.12 (ESRI, 2007) and Western Forest Complex (WEFCOM) (2004)’s topographic data were used to create the study area map. The altitude of the study site ranges from 700 m to 1,000 m above averaged sea level (asl). The average rainfall and temperature are 1,071 mm year−1 and 28 °C, respectively. The vegetation cover is mixed deciduous forest (60%), dry dipterocarp forest (30%) and disturbed areas (10%). The Lagerstroemia tomentosa, Terminalia alata, T. triptera, T. bellirica, and Afzelia xylocarpa are the dominant cover species in the habitat (Salakphra Wildlife Sanctuary, 2011).

Figure 1 Location of banteng (Bos javanicus) presence and camera stations in the Salakphra Wildlife Sanctuary.

This study was accomplished in compliance with the Department of National Parks, Wildlife and Plant Conservation (DNP 0909.204/ 7103), Thailand. A research ethics statement was granted by the Mahidol University-Institute Animal Care and Use Committee (MU-IACUC 2018/020).

Systematic reintroduction of banteng

Data were collected as previously described in Chaiyarat et al. (2019, 2020) as follows: Training of the banteng before reintroduction

During their time in captivity, the banteng underwent general medical checkups and received minimal human contact (Prakobphon, 1988; IUCN/SSC, 2013). Seven captive-purebred banteng were bred in a 302 ha enclosure. Four adult males and three adult females between five and seven years old were habituated with transportation boxes (1 m × 2.5 m × 1.8 m, width × long × high) individually for 6 months at the Khao Nampu Nature and Wildlife Education Center (Chaiyarat et al., 2019). They were then translocated to Salakphra Wildlife Sanctuary where they lived in a soft release cage (Sankar et al., 2013) for 4 months before release. In the soft release cage, they were kept in groups and their baseline BCS were assessed before being released. The captive-bred banteng had been raised on a diet of Zea mays Linn., Hymenachne pseudointerrupta C. Muell, Hewittia malabarica (L.) Suresh., Trichosanthes cucumerina L., fresh water and artificial salt licks. While in the training cage, the captive-bred banteng diet was switched to natural plants found in the cage. After reintroduction, natural food plants and salt-licks were the main food sources of the reintroduced banteng. These main food sources were increased according to the animals’ BCS and physiological states (Pokharel, Seshagiri & Sukumar, 2017).

Systematic reintroduction of banteng

Immobilizations of banteng were controlled with anesthetic drugs: (1) Thiafentanil oxalate 0.015 mg/kg (Thianil TM, Wildlife Pharmaceuticals (Pty) Ltd., Rocky Drift, White River, South Africa) and (2) Medetomidine HCl 0.015 mg/kg (Kyron Laboratories (Pty) Ltd., Benrose, Johannesburg, South Africa); and reversal drugs: (1) Naltrexone (Thianil TM, Wildlife pharmaceuticals (Pty) Ltd., Rocky Drift, White River, South Africa) and (2) Atipamizole HCl (Kyron Laboratories (Pty) Ltd., Benrose, Johannesburg, South Africa), by veterinarians of the National Parks, Wildlife and Plant Conservation and The Zoological Park Organization under the Royal Patronage of His Majesty the King. The animals were fitted with radio collars (<3% of body weight, very high frequency (VHF) transmitters; Advanced Telemetry Systems (ATS), Isanti, MN, USA) using standard capture and marking practices (Powell & Proulx, 2003) prior to transporting to Salakphra Wildlife Sanctuary. Radio collar signals were tested in the soft release cage before the banteng were reintroduced. First, collar signals were examined 1 week after reintroduction to reduce the bias when the banteng were not familiar with their new habitat. The radio-collared banteng were monitored periodically every week through ground tracking, using homing in and triangulation techniques (White & Garrot, 1990) via VHF signals (Chaiyarat et al., 2019). Four individuals of captive-bred banteng were reintroduced in December 2015 during the dry season (November–April). Shortly, three other individuals were reintroduced in July 2016 during the wet season (May and October).

Camera trap survey

Cameras traps (Bushnell 12 MP Trophy Cam HD Essential Trail Camera, Suresnes, France) were installed between 2016 and 2018 after the second group of banteng were reintroduced. After the camera traps were installed, memory cards and batteries were changed every month in each location for the entire 3 years. Camera trap locations were selected based on a radio transmitter survey to reduce bias due to trails and other features (Kolowski & Forrester, 2017). Therefore, water sources within the study site were primarily used as locations for camera trap placement (Varma, Pittet & Jamadagni, 2006) followed by natural licks and wildlife trails (Rovero & Marshall, 2009), which are often visited by banteng and other large mammals (Chaiyarat, Youngpoy & Prempree, 2015; Chaiyarat et al., 2019) such as wild Asian elephant (Elephas maximus), gaur (Bos gaurus) and sambar deer (Rusa unicolor), etc. Each trap station was installed with two cameras opposite to each other, positioned to photograph both asymmetrical flanks of the banteng for positive identification (Soisalo & Cavalcanti, 2006). Camera traps were mounted on trees at about 0.75 m height above ground (Rowcliffe et al., 2008). Camera traps were installed at points ranging between 1 and 3 km apart. The camera traps operated continuously, 24 h per day and the camera shooting interval was 1 min. The pictures had a resolution of 1,648 × 1,236 pixels. Camera ID, time, date and temperature were also recorded for each exposure and were stamped on the photographs (Chaiyarat et al., 2019). A total of 1,879 photographs, out of 191,748 photographs recorded by 12 of 32 camera trap locations (Fig. 2) during 4,602 trap-nights were identified as having captured the banteng (Chaiyarat et al., 2019).

Figure 2 The body composition for body condition scoring (BCS) of banteng (Bos javanicus) (A) and photographic features of banteng in each location of body composition in each BCS criteria (B) in the Salakphra Wildlife Sanctuary.

Body condition scoring

The morphological characteristics: scars on the body, shape of horns and forehead, labeled collars and pairing of mother-calf were used for identifying and referring to the specific individuals. Only sufficiently visible images which showed the whole body of the animal were selected for scoring. The visual body condition of banteng was scored by a five-points pictorial scoring system developed from scoring indexes of Bornean banteng (Prosser et al., 2016) and Bali cattle (Soares & Dryden, 2011). Each individual was given a score in the appearance of seven key areas of its body. Following the index criteria (Table S1; Fig. 2), the neck (male = 17 photos, female = 18 photos), dewlap (male = 18 photos, female = 18 photos), shoulders (male = 18 photos, female = 17 photos), vertebrae (male = 18 photos, female = 16 photos), ribs (male = 18 photos, female = 18 photos), hindquarters (male = 18 photos, female = 17 photos) and tail head (male = 13 photos, female = 11 photos) of the seven bantengs’ body (n = 36 photos, 18 photos for both males and females) were categorized with prominent skeletal features and soft tissue by the five-points scoring system. The body composition scores for each area were tabulated to create a database for analysis and future surveys. Body condition score values were tabulated by three interrelated-assessors: graduate students who developed the BSC (Assessor A), a banteng specialist who contributed to the development of the banteng BCS and has worked with banteng for more than 20 years (Assessor B) and a veterinarian with no prior experience in BCS of any species (Assessor C) (Wemmer et al., 2006; Morfeld et al., 2016). After training, the assessors were tested for consistency using a pilot study of captured photographs obtained from ten captive banteng which were then reanalyzed as recommended by Pokharel, Seshagiri & Sukumar (2017). The level of percentage agreement was >80% as an almost perfect agreement before processing as recommended by Landis & Koch (1977) and Morfeld et al. (2016). In camera-traps, lighting effect, moving animals, clarity and many other factors might complicate the visualization of fat depositions, depressions and projections of bones, etc. These were reanalyzed by comparing with the photographs of captive individuals.

Statistical analysis

The camera-traps were used to capture photographs continuously over the course of the study. The Spearman’s rank correlation between categorical body composition and BCS of the four male and three female banteng were used to assess changes in their condition (Wayne, 1990) by using the R package (Niedballa et al., 2016). Chi-square was used to compare the BCS of the banteng between categorical factors such as wet and dry seasons. Mann–Whitney–Wilcoxon rank of the BCS categories from the first, second and third year after the reintroduction of the banteng were determined by using SPSS (Mann & Whitney, 1947). The differences between treatments were tested at the p < 0.05 significance level.

Results

Body condition scoring system

A total of nine banteng composed of three bulls, four cows and two calves were captured by camera traps. Only the seven adult banteng were used to investigate BCS. The capture rates of banteng photos were 30.8% of all photos in the first year, 77.8% in the second and 20% in the third year. In this study, the criteria for BCS were based on the body composition of seven areas. The Spearman’s rank correlation between body composition and BCS of both reintroduced male and female banteng were not different, except in neck and dewlap (male and females), neck and shoulder (males and females), neck and vertebrae (females), neck and rib (males and females), neck and hindquarter (females), dewlap and shoulder (males and females), dewlap and vertebrae (males), dewlap and ribs (males and females), dewlap and hindquarter (males and females), dewlap and tail head (females), vertebrae and ribs (females), vertebrae and hindquarter (males) and hindquarter and tail head (females) that were significantly different (p < 0.05) (Table 1).

Table 1 Spearman’s rank correlation of body key areas’ score of the reintroduced banteng (Bos javanicus) in the Salakphra Wildlife Sanctuary.

	Body key areas (Photos)	Neck
(n = 17)	Dewlap
(n = 18)	Shoulder
(n = 18)	Vertebrae
(n = 18)	Rib
(n = 18)	Hindquarter
(n = 18)	Tail head
(n = 13)		
Female (n = 3 individuals)	Neck
(n = 18)		0.51**
(0.001)	0.71**
(<0.001)	0.24
(0.084)	0.53**
(0.001)	0.25
(0.068)	−0.26
(0.239)	Male (n = 4 individuals)	
Dewlap
(n = 18)	0.57**
(<0.001)		0.68**
(<0.001)	−0.52**
(0.001)	0.58**
(<0.001)	−0.51**
(0.001)	0.16
(0.450)	
Shoulder
(n = 17)	0.55**
(<0.001)	0.70**
(<0.001)		−0.29*
(0.042)	0.36*
(0.016)	−0.25
(0.068)	0.29
(0.186)	
Vertebrae
(n = 16)	0.77**
(<0.001)	0.12
(0.297)	−0.10
(0.380)		−0.1
(0.445)	1**
(<0.001)	0.52*
(0.029)	
Rib
(n = 18)	0.39**
(<0.001)	0.52**
(<0.001)	−0.11
(0.259)	0.53**
(<0.001)		−0.09
(0.485)	0.29
(0.186)	
Hindquarter
(n = 17)	0.32**
(0.007)	0.80**
(<0.001)	0.50**
(<0.001)	0
(1.000)	0.22
(0.052)		0.52*
(0.029)	
Tail head
(n = 11)	0.19
(0.292)	−0.55**
(0.008)	−0.26
(0.160)	0.40*
(0.040)	−0.16
(0.356)	−0.73**
(<0.001)		
Notes:

* p ≤ 0.05.

** p ≤ 0.01.

The mean BCS for the body regions of reintroduced banteng ranged between first and fifth level (Fig. 3; Table 2). In the first year after reintroduction, BCS of females were ranked at 2.43 ± 0.41 and skeleton features were still recognized. While the BCS of males increased to 4.35 ± 0.34, and all of the skeletal features were covered with soft tissue in the third year (Fig. 3).

Figure 3 The body condition score (BCS) improvement of three males and four females banteng (Bos javanicus) between wet (males = four photos individual−1, females = three photos individual−1) and dry (males = two photos individual−1, females = three photos individual−1) and dry (males = two photos individual−1, females = one photo individual−1 for two individuals and two photos individual−1 for two individuals) seasons in the Salakphra Wildlife Sanctuary after the reintroduction.

Table 2 Body condition score (BCS), chi-square and Mann–Whitney−Wilcoxon rank test of male and female reintroduced banteng (Bos javanicus) between the first and the third year of reintroduction program in the Salakphra Wildlife Sanctuary.

Sex	Body condition score in season and year after reintroduction	
Wet	n	Dry	n	1st	n	2nd	n	3rd	n	
Male	3.37 ± 0.78	12	3.53 ± 0.50	6	2.98 ± 0.35	77	3.59 ± 0.62	30	4.35 ± 0.34	30	
Female	2.83 ± 0.39	12	2.67 ± 0.52	6	2.43 ± 0.41	94	3.14 ± 0.30	22	3.21 ± 0.10	16	
Both sexes	2.95 ± 0.78	24	3.19 ± 0.53	12	2.67 ± 0.47	171	3.40 ± 0.53	52	3.97 ± 0.64	46	
Compared rank	Season	df	χ2	p-value	Year	N	Z	U	p-value	
Male	Wet vs. Dry	1	0.882	0.348	1st year vs. 2nd year	30	−2.72	25.0	0.006**	
Female	Wet vs. Dry	1	0.607	0.436	2nd year vs. 3rd year	13	−1.50	10.5	0.133	
Both sexes	Wet vs. Dry	1	2.202	0.138	1st year vs. 3rd year	29	−3.29	8.0	0.001**	
Note:

** p ≤ 0.01.

Improvement of body condition scoring after reintroduction

Overall, the BCS of the reintroduced banteng were not significantly different (p > 0.05) between seasons in both males and females, but the BCS of males were higher in the dry season (Table 2; Fig. 2), but it significantly increased over time in an upward trend after they were reintroduced into the natural habitat, especially in the oldest male (PM) (Table S2; Figs. 3–5). The average BCS of reintroduced banteng in males was higher than females in all time periods (Table 2). In males, the BCS increased dramatically between the first (x¯ = 2.98 ± 0.35) and third years (x¯ = 4.35 ± 0.34, p < 0.05), while in females, BCS were slightly steady between the second (x¯ = 2.43 ± 0.41) and third years (x¯ = 3.21 ± 0.10, p > 0.05). The average BCS of both sexes gradually rose from the first year (x¯ = 2.47 ± 0.64) to the second year (x¯ = 3.40 ± 0.53) and the third year (x¯ = 3.97 ± 0.64), respectively. The Mann–Whitney U tests showed that the BCS of banteng in both sexes were different between the first and second years (Z = −2.72, U = 250, p = 0.006, n = 30) and between the first and third years (Z = −3.29, U = 80, p = 0.001, n = 29) (Table 2).

Figure 4 The body condition score (BCS) improvement of four females (open dots) and three males (closed dots) banteng (Bos javanicus) in the Salakphra Wildlife Sanctuary in the first (n = 13 photos of three females and n = 10 photos of three males), second (n = three photos of two females and n = four photos of two males) and third year (n = two photos of one female and n = four photos of two males) after the reintroduction.

Figure 5 Rate of increase in body condition score (BCS) of banteng (Bos javanicus) in females compared to males in the Salakphra Wildlife Sanctuary in the first (n = 13 photos in three females and n = 10 photos in three males), second (n = three photos in two females and n = four photos in two males) and third year (n = two photos in one females and n = four photos in two males) after the reintroduction.

Discussion

In the Salakphra Wildlife Sanctuary, the BCS of banteng increased in both males and females after being reintroduced into a natural habitat by using camera-traps. The BCS were not significantly different between wet and dry seasons. The average BCS of male banteng increased in the dry season and decreased in the wet season because the resource quantity and quality in the wet season were higher than in dry season and the early wet season. As the small sample size and individual differenced in BCS changed over the study period, they may influence the results.

Monitoring of BCS by using camera-trap is noninvasive to the animals and highly effective in the field due to the low cost and time consumption. The camera-traps can be installed over large areas and can take photographs of animals continuously for long periods of times. As the method is a subjective assessment based on visual appraisal and comparison, there is bound to be variation caused by lighting, posture and observer. However, the error was found to be comparatively small (Fernando et al., 2009). The visual features of each body compositions captured by camera trap can be used to investigate the development of BCS in banteng. The BCS assessment was previously recommended for use to investigate Bornean banteng (Bos javanicus lowi) in logging forests in Sabah, Malaysia (Prosser et al., 2016).

The BCS system itself is a robust measure. The results indicated that most of the key areas assessed were strongly correlated, which suggests that the scoring scheme could be simplified without losing much information (Zielke, Wrage-Mönnig & Müller, 2018). This study found that dewlaps, ribs and hindquarters gave significantly different results from other body key areas. Nonetheless, more data points of the reference are still preferred. The potential of the reference features should be maximized when a short series of photographs are scored (Prosser et al., 2016). Meanwhile, the quality of the photographs including brightness and animals’ posture might result in differential scoring of each structure. Skeletal features are clearly visible when an animal is stretched out mid-stride but less visible when its posture is relaxed which affects the perceived body condition (Wemmer et al., 2006).

The correlations among the scoring of key body areas were assumed to be not significantly different. In this case, all of the seven key areas of the banteng’s body significantly correlated to at least two other areas. These findings call attention to the significance of monitoring banteng BCS and also offers the possibility for estimating parameters in relation to the various body compositions scoring when some key area is missing (Zielke, Wrage-Mönnig & Müller, 2018). Nevertheless, validation and calibration of the BCS scoring are still required to decrease the bias of an estimator by assigning the BCS on the training cage and captured photographs, which was then reanalyzed by the observer and other assessors (Pokharel, Seshagiri & Sukumar, 2017).

After reintroduction, the BCS in all individuals increased in the first year. The higher trend of BCS development in bulls may have been affected by their biological traits. High levels of testosterone in males can influence their anatomical and behavioral features, for instance, the large body size, rapidly bulking up and expressing more aggression, especially during the mating period (Hinch & Thwaites, 1984). Males tend to develop their body condition more than females, especially during reproductive periods to compete with other males, for reproductive benefit more than females. On the other hand, the steady improvement of the females’ body condition improvement may be the effect by parturition and lactation during their maturity period similar to dairy cows (López-Gatius, Yániz & Madriles-Helm, 2003) and elephants (Pokharel, Seshagiri & Sukumar, 2017, 2020) as two calves were found in the third year.

The fluctuations of BCS can reflect the quality and quantity of resources and other factors such as parasites or even mortality (Dube, 2005). In order to develop the efficiency of banteng body condition visual scoring in monitoring the reintroduction program, broader details about growth factors, such as age, genetic traits and nutritional status of captive-bred banteng before and after reintroduction should be implemented in different situations of habitat type, seasonality and stress levels, by comparing across gradients of fragmentation and anthropogenic disturbances as recommended by Pokharel, Seshagiri & Sukumar (2017). This could contribute to planning for the welfare, conservation and management of reintroduced banteng populations.

Conclusions

This study has presented a practical technique to evaluate the BCS of reintroduced banteng. The indirect observation of banteng by using the photos from camera traps can be applied. The monitoring allows us to stay connected to the animals responsibly to implement future program management, either by taking active plans or verifying the program’s effectiveness. Although a large number of recaptured photographs of animals and a series of individual profiles are required, the integration of visual body condition scoring with photographs of camera trapping is a good option for monitoring the establishment of a small population. This finding also reveals the positive trend of the banteng reintroduction program as a conservation management tool by humans. Continued monitoring and validation are required for this program implementation to act as a future guideline for other reintroduction programs. The BCS systems should be considered for monitoring reintroduced banteng in other areas and further evaluations are recommended, including correlations of BCS with body lipid content, subcutaneous fat, weights, shoulder heights and any other fitness-related traits.

Supplemental Information

Supplemental Information 1 Raw data.

Click here for additional data file.

Supplemental Information 2 The scoring criteria for seven body compositions of the banteng (Bos javanicus) in Salakphra Wildlife Sanctuary.

Table S1 The scoring criteria for seven body compositions of the banteng (Bos javanicus) in Salakphra Wildlife Sanctuary.

Click here for additional data file.

Supplemental Information 3 The seven body compositions of the banteng (Bos javanicus) in each individual, sex, age, year of reintroduction in Salakphra Wildlife Sanctuary.

Table S2 The seven body compositions of the banteng (Bos javanicus) in each individual, sex, age, year of reintroduction in Salakphra Wildlife Sanctuary.

Click here for additional data file.

We thank the Salakphra Wildlife Sanctuary and Khao Nampu Nature and Wildlife Education Center, Department of National Parks, Wildlife and Plant Conservation and the Zoological Park Organization Thailand for providing permission and assistance for data collection. Our appreciation and thanks to Dr. Thomas N. Stewart (USA), Mahidol University, Thailand, for editing the manuscript.

Additional Information and Declarations

Competing Interests

Author Contributions

Animal Ethics

Field Study Permissions

Data Availability

The authors declare that they have no competing interests.

Praeploy Kongsurakan conceived and designed the experiments, analyzed the data, prepared figures and/or tables, authored or reviewed drafts of the paper, and approved the final draft.

Rattanawat Chaiyarat conceived and designed the experiments, performed the experiments, analyzed the data, prepared figures and/or tables, authored or reviewed drafts of the paper, and approved the final draft.

Seree Nakbun performed the experiments, authored or reviewed drafts of the paper, head of the field work, and approved the final draft.

Nikorn Thongthip conceived and designed the experiments, authored or reviewed drafts of the paper, and approved the final draft.

Panat Anuracpreeda conceived and designed the experiments, authored or reviewed drafts of the paper, and approved the final draft.

The following information was supplied relating to ethical approvals (i.e., approving body and any reference numbers):

Mahidol University-Institute Animal Care and Use Committee provided full approval for this research (MU-IACUC 2018/020).

The following information was supplied relating to field study approvals (i.e., approving body and any reference numbers):

The Department of National Parks, Wildlife and Plant Conservation, Thailand approved field collection (DNP 0909.204/ 7103).

The following information was supplied regarding data availability:

Raw data is available in the Supplemental Files.

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
