# Peer review of "Monitoring body condition score of reintroduced banteng (Bos javanicus D’Alton, 1923) into Salakphra Wildlife Sanctuary, Thailand"

_PeerJ, doi:10.7717/peerj.9041_

## Round 0.1 · original submission · Major Revisions

As you can see, all three of my reviewers want you to make extensive changes to your manuscript. Unless, I have misjudged them, I think all these changes are fairly simple. The reviewers have been particularly detailed so I believe it will be straightforward for you to correct your work accordingly.

In your response, please be explicit in how you have addressed their concerns.

Reviewer 1 ·

Basic reporting

The manuscript “Body condition scoring system of reintroduced banteng (Bos javanicus D’Alton, 1923) into Salakphra Wildlife Sanctuary, Thailand” was well written and showed the reintroduction process of large herbivore as Banteng. The advantage of this manuscript is the work with reintroduction of Banteng, which is unique work. Moreover, the authors analyzed the correlation of the body condition scoring index. However, the low number of animals in the findings was low, compared to the previous work (9 Banteng in this study compared to 111 bantengs in Prosser, N S et al. (2016).

The topic should be changed to
“Monitoring Body condition score of reintroduced banteng (Bos javanicus D’Alton, 1923) into Salakphra Wildlife Sanctuary, Thailand” as the authors did not develop the body condition scoring system, but rather used the method developed by Soares, F. & Dryden, G.M. (2011) and Prosser, N S et al. (2016), even camera trap was used which was similar to the work by Prosser, N S et al. (2016)

The English should be rechecked as typing and grammatic errors, which could be better and fluent English.

The authors took three bulls, four cows and two calves into the calculation. I don’t know that the BCS of calves were included in the calculation or not, which probably was not appropriated. The authors showed the different between male and female, but did not show the different between adults and calves.

Table 1 could be put in the supplement

Experimental design

The use of camera trap to identify the banteng BCS as well as method for identify individuals was good, and well design, as described for the objectives of the study.
Please add more on the statistical analysis for one more paragraph.

Validity of the findings

The findings were good with the results compared from year by year of reintroduction.
However, BCS in Banteng were not novel findings, the advantage of this study was newly performed in the reintroduction group, not a wild population.

Additional comments

Line 82-83
“However, they are protected under the Reserved and Protected Animals Act, B.C.2535 of Thailand (Lekagul & McNeely, 1977).”
This is weird to refer to this article, as this act was announced in BC2535 (1992), but the authors referred to the article in 1977, which was 15 years earlier than the act was announced.

Line 93
The wild banteng population in Thailand was estimated at only c.470 in the 1990s

What is “c.470” This is the number? If it is specific term in this field, please explain more for the readers who are not in this field.

Line 164-168
Were there any other large mammals e.g. guar, any kind of deers, wild buffalo etc.
Banteng was mentioned 2 times in this sentence, please remove one.

Line 168-169
This phrase and references “banteng (Chaiyarat et al., 2019; in 30 m 169 UTM pixel, ArcGIS 9.3, ESRI 2007)” were mentioned here, which was not related to the context.
About the camera trap or methods should be mentioned elsewhere.

Line 169 and 320
ESRI (2007) and
ESRI. (2007). ESRI® Data & Maps 2006. NY: ESRI
Should be clarified more, as the readers may not understand the meanings, or any context why this reference is here. What does ESRI stand for?

Line 202
“BSC” should be changed to “BCS”

Line 202
The authors can use abbreviation “BCS”

Line 273
Please correct “are required,t he integration”

These references are not listed in the text, even they are the statistical analysis papers, it should be referred somewhere else.

Corder, G.W. & Foreman, D.I. (2014). Nonparametric Statistics: A Step-by-Step
Approach. Hoboken, NJ: Wiley.
Mann, H.B. Whitney, D.R. (1947). On a Test of Whether one of Two Random Variables
is Stochastically Larger than the Other. Ann. Math. Statist. 18(1), 50-60.
Niedballa, J., Sollmann, R., Courtiol, A. & Wilting, A. (2016). camtrapR: an R package
for efficient camera trap datamanagement. Methods Ecol. Evol. 7, 1457–1462.
Wayne, W.D. (1990). Spearman Rank Correlation Coefficient. In Applied Nonparametric
Statistics: pp. 358–365. 2nd edn. Boston: PWS-Kent.

·

Basic reporting

The manuscript “Body condition scoring system of reintroduced banteng (Bos javanicus D’Alton, 1923) into Salakphra Wildlife Sanctuary, Thailand” addresses the important aspect of conservation management, i.e., the effect of reintroduction on banteng. There are very few publications on this species, so the authors’ efforts of studying such animals should be commended. The authors have advanced the conventional way of measuring body condition by the camera-traps. The manuscript is well thought of. However, there are several major aspects which I was concerned about, particularly related to writing style, study design and analyses. Thorough proofreading of the manuscript is recommended. I have listed these concerns, as per the Journal format, below:

Basic reporting:
1. The title “Body condition scoring system of reintroduced banteng (Bos javanicus D’Alton, 1923) into Salakphra Wildlife Sanctuary, Thailand” needs modification. It is not clear whether the authors are trying to highlight the body condition scoring system? Or, the effect of reintroduction on banteng’s physical health?
2. Abstract:
a. Background: Lines 31-33 seem very abrupt. It is also important to highlight briefly why such studies on ‘banteng’, in particular, are important. A brief statement about their conservation status would be valuable.
b. Methods: The authors should state how they score the body condition. It was not direct visual scoring, but the photographs through camera-traps. This aspect should be included. It appears that only “Two newborn banteng” were captured via camera traps in 2018 from the first group that was reintroduced in 2015. This should be explicitly explained.
c. Line 41: …into the natural habitat.
d. Results: What were the age-differences? Were they adults? Or, a newborn? Was the increment statistically significant? A clearer statement is recommended.
e. Conclusions: This needs to be very strong. The current statements under the Conclusions “The results from the present study suggest that the BCS of reintroduced banteng can be used to monitor the body condition of reintroduced banteng. These findings are important for possible translocation elsewhere.” should be modified. Lines 45-46 appears repetitive, rather format it as the body condition of reintroduced banteng can be used to monitor “the effect of reintroduction…” And how these findings are linked with translocation?
3. Line 56: clear space before (IUCN).
4. Lines 67-77: How is the social dynamics/grouping in banteng? The authors highlight the global population of banteng, then Line 95: they mention the local population, and state that it is increasing, so what is the current population now? They should mention the current status of banteng.
5. Lines 113-114: Morfeld et al., 2016 was on captive Asian elephants. Authors can also refer the study on wild populations of Asian elephants: Pokharel et al., 2017 (Conservation physiology) studied the effect of seasonality (variations in resources) on body condition and stress of wild Asian elephants.
6. Lines 201 to 202: Font colour appears different.
7. Lines 205 to 215: Kindly mention the sample size here and wherever required.
8. Line 273: correct “the”
9. Line 274: No “full stop”
10. Scientific names to be in italic, see lines 300, 303, 314, 345, 357, 363, 371, 374, 375, and wherever required.
11. All table legends: Banteng scientific name to be in italic and same applies for Figure 1. Figure 2 and others.
12. Figure 2: It’s BCS, not BSC.
13. Legends are in different fonts, make it uniform.
14. Thorough proof-reading is recommended.

Experimental design

Experimental design:



1. Lines 116 to 118 should clearly state the hypothesis and objectives (though have been included in the current manuscript) of this study. It is important to justify how studying the body condition will promote the conservation of this bovid species (?). The study should be hypothesis-driven.
2. Methods need major corrections. Issues pertaining to methods/designs are listed below:
a. If the studied animals were monitored using the radio-collared signals over a period of time and also photographed using camera-traps, why was not seasonality included in the analysis?
b. As seasonality (as a proxy of resource distribution) has been shown to influence the body condition and also animals’ physiological states (Pokharel et al., 2017), the authors should define the change in body condition in terms of resources. The influence of dietary shift between captivity and the reintroduced sites cannot be discounted.
c. Age-class wise variation should also be included.
d. The authors have not explained the statistical analyses. This should be added. Why were Spearman’s correlation and Mann Whitney U/Mann Whitney Wilcoxon tests performed? Based on what these tests were selected? What was the distribution of the data? What software was used to analyze? What were the sample sizes for male, female and their age classes? All details should be included.
3. Were the scoring criteria defined in Table 1 designed by the authors for their study? If so, kindly mention this. If it was adopted from other studies then cite them.

Validity of the findings

Validity of the findings:


1. Though the authors used camera-traps for assigning the body condition score, the authors should/must validate that the use of camera-trap photographs for scoring the body condition. Even in the normal scenarios, where photographs are captured for assigning the body condition, the scores can be highly biased as the animal might be in different postures: moving, not on the uniform ground, angle of light exposure, our distance from an animal, etc. This might affect the scores. Thus, most of the research validate their studies by including inter-assessor variations. Pokharel et al. (2017) validated the scores by assigning the BCS on the field itself and also capturing the photographs, which was then re-analyzed by the observer and other assessors. In camera-traps, lighting effect, moving animals, clarity and many other factors might complicate to visualize the fat deposition, depression and projections of bones, etc (Figure 2 is the right example for such caveats). It is thus highly recommended to either validate the camera-trap photographs with normal photographs or to refer to the published work where they have used a similar technique and validated the scoring system.
2. Line 198 onwards: Body condition score system: would not cropping an image change the angle of deposition or shadow? Would not that bias the scoring? Validation or proof of validation of the camera-trap technique is highly recommended.
3. Lines 222 to 223: A graph to visualize this trend is recommended. Kindly refer to Figure 3.
4. Lines 225 to 227: terms ‘dramatically and slightly steady’ should be backed up with p-values.
5. Graphical visualization of change in patterns of body condition is highly recommended. Figure 3, in this context, should be slightly modified. Does the red or blue dots represent female and male? Does that mean there were 13 females in the first year? How many photographs were assessed per individual in a particular year?
6. The discussion should highlight the possible reasons behind increasing body condition and its implication on evaluating the effectiveness of the reintroduction. And how this is important for the conservation of banteng.

Reviewer 3 ·

Basic reporting

This manuscript is well written, but there are a few areas that could be rearranged to improve flow, and a couple of sentences where I would appreciate re-wording or clarification.
I also suggest the authors add an 'analyses' section to the methods to clarify exactly how the data were statistically analysed.
The raw data for how each BCS score was calculated from the seven body areas is included, but does not indicate which individual was assessment, so it is difficult to assess how over-representation of certain individuals may bias the results (i.e. 36 assessments from 9 individuals). I think this is important information to include.

Experimental design

The authors should more explicitly outline their goals in the introduction, and add an analyses section to outline what they did - this would help the reader interpret the results, and would allow for replication as required.
More information about how the data were handled would also make it easier to assess whether appropriate statistical analyses were performed.

Validity of the findings

More information is needed on how many of the 36 data points were contributed by each of the nine individuals - it is difficult to assess whether the results are statistically sound without this.
Clarification on the potential influence of season is also required - if BCS was assessed in the wet or dry season, or a combination, would that impact the results?

Additional comments

This study uses camera trap photographs to repeatedly assess the body condition of a small population of reintroduced banteng, as a method of assessing their acclimatization over a period of three years. This is a great idea that could also be applied more widely across other populations and species where individuals are identifiable from camera trap photos. There are a few areas where I would like some clarification (such as whether there may be any seasonal effects that should be considered here, and how exactly the data points were created), and I feel the statistics may need to be reassessed to take into account season and the repeated observations across individuals, but overall I think this is a useful study and should be considered for publication, pending revision.

My main concern with the manuscript as it stands, is that I don’t understand exactly how the BCS data were created, so I can’t assess the suitability of the statistics used. It would be helpful if you could explain exactly how the data were used in your analyses – you had 376 images, which ended up as 36 data points, but only a maximum of 9 individuals each year (only 7 of which were adults). I understand that individuals were measured across the three years, but there must be multiple measurements for some individuals within a year also – such as year 1 where you have 13 female and 10 male data points, but no more than 7 adults. If the BCS scores were taken at different times throughout that year, then this needs to be explained. I am also unclear on where the two calves fit into the data – I feel it would be unusual to compare the BCS of a calf with that of an adult, so this needs to be clarified. I am also slightly concerned that since there were fewer data points in years 2 and 3, whether there are enough to assess the between year comparisons statistically. I think this information should be made more clear in the manuscript itself, rather than just if readers look at the supplemental material.

Please add a section to the methods outlining the statistical analyses that were performed, including how the photographs were used to create your BCS scores (i.e. how many assessments per year per individual, whether these were averaged by year or used as discrete measurements), the variables and statistical tests used. I do have some concerns over the statistical analyses performed - you need to take into account in your analyses the potential confound of season, and your analyses should also take into account that you have repeated measures from the same individuals over time (even within a year perhaps) – this has the potential to bias your results if some individuals are over-represented, but there are options for using all the data but taking these things into account statistically. It also seems as though the rate of BCS change over time differed between males and females – this should be considered within the analyses if your sample size allows, or discussed further if it doesn’t.

A couple of specific questions relating to this:
The authors state that seven banteng were reintroduced in 2015 (four individuals) and 2016 (three individuals) (lines 37-38), and that camera traps were set up for three years following release (line 183), during fieldwork from January 2015 to November 2017 (line 35). How far apart were the two releases, and does this mean the 2016 individuals were only assessed twice (in 2016 and 2017), or that the 2015 release individuals had been out longer than the 2016 individuals when first assessed? Please could you clarify this information?

I am unfamiliar with the seasonal conditions in this area (although I note the authors mention a dry and a wet season (lines 97-98) – please could you elaborate on this a little, and explain whether the time of year that the BCS was assessed was the same for the three years of assessment (and for the potential multiple assessments within a year), or if not, whether there is any reason to suspect that the different rate of BCS change between years 1 to 2 and 2 to 3 may be confounded with any seasonal effects.

I have also added some comments/suggestions by section/line below.

Abstract:
Line 30: I suggest that you add quality to this first sentence – ‘Large forest-dwelling mammals are highly sensitive to habitat structure and quality’, because I think BCS is going to be reflective of the quality of the habitat.
Lines 43-44: I don’t see the female BCS decrease during lactation in the results – please could you expand on this in the results, or rephrase here.
Lines 45-47: I suggest editing this section to something like: ‘The results from the present study suggest that camera traps are a practical tool to assess the BCS of reintroduced banteng, and can be used to monitor their condition post-release. These techniques are important for possible translocation programs elsewhere.’

Introduction:
Lines 60-61: Is there a word missing from this sentence – ‘common conservation strategy’ perhaps?
Line 62: I suggest ‘including the endangered banteng’ here
Lines 63-64: I suggest adding another sentence here explaining the situation - introduction into a zoo doesn't sound like a reintroduction, so was this into a semi-wild situation, or what exactly happened? And also, what criteria do you use to say this was successful – has the population grown since reintroduction?
Lines 65-66: I suggest that you add a sentence here to explain that this introduction is the one that is being used for this study – I didn’t make that connection without looking up the reference, so it would help the reader to include that information more explicitly here.
I would suggest reorganizing the introduction slightly to help guide the reader through the issue you are addressing with your study. I would perhaps move lines 63-66 to after line 96 ‘In Salakphra Wildlife Sanctuary, banteng were locally extinct’. This would set up the importance of the banteng as an endangered species, explain the threats and low population size, and then discuss the reintroductions that have been done so far and the need to assess BCS because of the shift in diet.
Lines 80-81: I’m not sure that I understand your statement ‘since banteng that are traded in the market are a domestic species and not a wildlife species’ – do you mean that these are hybridized with domestic cattle, or something else – please could you clarify this?
Lines 89-90: I suggest reordering this sentence: ‘Accidental hybridization in the wild and active inbreeding between banteng and domestic cattle in captivity to develop the livestock industry reduces the purity of the genetic status of both the wild and captive populations’
Lines 116-117: It would be helpful if you elaborated a little more on your specific aims here – that you assessed BCS over three years using camera trap photos, that way the reader knows what your specific goals were before they get to the results.

Methods:
In the abstract you mention that the reintroductions were done at two times - 2015 (four individuals) and 2016 (three individuals) – please could you add this to the methods, and explain why these were done in two sets, and how far apart the releases were done as this may influence the BCS data if the time between release and the first camera trap photos was very different for the two groups (especially since they were released in the dry and wet season, respectively, as per lines 97-98).
Lines 126-129: Do these species make up a large proportion of the banteng diet? Please expand on why this information is important for the reader to understand
Line 139: What training did they receive?
Line 142: Were the soft release cages also individual, or were they kept in groups?
Line 143: Was there any opportunity to assess their BCS before the release?
Line 145: Do you mean tap water? I might re-word to ‘provided with fresh water and artificial salt licks’
Line 149: What were the banteng immobilized with?
Line 163: This should read ‘to reduce bias’
Line 168-169: I suggest moving the reference/information to the previous line after ‘banteng’ and before ‘and other large mammals’: (Chaiyarat et al., 2019; in 30 m 169 UTM pixel, ArcGIS 9.3, ESRI 2007) to improve flow.
Line 169-171: I would move this sentence on the number of photographs to the end of this section, after you have described the set-up of the cameras.
Line 175: I recommend deleting ‘along each stretch’ and leaving this as ‘camera traps were set up at distances ranging between….’
Line 183: I would move this to the section on camera trap surveys, along with the other camera set up methodology. Also, this should read ‘Camera traps were set up’
Line 183: Please expand on the details here – were the cameras set up for the entire 3-year period, or only for certain times across those three years? If only for certain times, was this during the wet season, dry season or a combination of the two? This could influence the BCS, and so is relevant information for the reader to understand.
Line 186-187: You state that ‘Photographic exposure and sharpness were adjusted due to poor quality visualization’ – was this just where necessary, or were all photos modified?
Line 187: This should read ‘sufficiently’
Lines 188-189: This might read more clearly as ‘The visual body condition of banteng was scored on a five-point pictorial scoring system’
Lines 193-194: I suggest moving this sentence up to before the previous one – state that it is seven region scoring system, then list those regions
Line 194-195: The following might read more clearly: ‘The body composition scores for each area were tabulated to create the database for analysis and future survey’

Results:
Line 201: Please could you explain what you mean by capture rates here – do you mean you only photographed 30.8% of the target animals in the first year, 77.8% in the second and 20% in the third year? How many repeated photos of the same individuals did you obtain, and how were these spread out temporally? It is important for the reader to understand where the data you use in your analyses came from – how many individuals were represented, and if the same individual was re-scored, was this a week apart or 6 months.
Lines 202-203: I suggest rephrasing to ‘criteria for BCS were based on body composition of seven areas’ (also note you had BSC here, instead of BCS)
Lines 203-24: You had not introduced what statistical comparisons you were making previously, so please ensure that information is added to the methods, as it is difficult for me to assess your results when it is not clear how data were processed/analysed.
If you were using each photograph, Spearman’s rank correlation would not be an appropriate analysis to perform here – my understanding is that you are looking to see how the BCS score of one region correlates with the BCS score of each other region, however you have repeated observations of the same individual multiple times, so your analyses could be biased by inequality in the number of photographs of the same individual. If you had, for example, an individual with poor correlation between two body parts, and that individual happened to be photographed more times than another where the body parts correlated better, then your overall result will be affected by this pseudoreplication. However, based on the degrees of freedom reported, this does not seem to be what you have done (although the df doesn’t match the number of individuals either – so perhaps mean BCS for each individual sighting event within each year, which would also have an element of pseudoreplication?). Please clarify how the number of photographs translated in data points and how this analysis was performed.
Lines 216-219: Again, please clarify which data were used to calculate these averages – was each individual included multiple times within each year? If so, this needs to be taken into account in your analyses. Also, on Figure 3, I don’t see any data points that scored a BCS 5 (and the lowest point was <2) – so do you mean that scores for the body regions ranged between 2 and 5? Please could you clarify this.
Line 223-229: Since there was a significant difference in BCS between males and the females, these analyses would be more appropriate if separated by sex (or sex used as an interaction term). I would suggest that your best approach, taking into account my questions regarding season and repeated assessments of the same individuals, is to use a mixed effect model. This would allow you to use repeated measures from each individual (within a year as well as between years), and you could also incorporate season and sex as covariates. This should strengthen your analyses, as you will be able to determine if males increase in BCS faster than females (which it looks like they may do). Additionally, it might be interesting to look at the female(s) that reproduced separately from those that did not – even if by separating individuals out in Figure 3 if you cannot do it statistically, so that we can see whether the slower rate of increase in BCS in females compared to males is confounded by the fact that two of the females reproduced and were lactating, reducing their BCS relative to the female(s) that did not breed.

Discussion:
Lines 235-237: I’m not sure if I understand this sentence – do you mean that the BCS assessment that you used here was previously recommended for use in Sabah? I suggest re-wording this sentence.
Line 245: I suggest you delete ‘reintroduced banteng’ from this sentence – the same would be true of any species, so by keeping this statement general, you increase the relevance to readers interested in using this approach for other species.
Line 249: I’m not sure that I understand your meaning in this sentence - could you please clarify?
Line 253-254 and 264-265: these are excellent points - could you suggest ways in which the method could be validated and calibrated? If you have access to the captive-bred animals, this is something that could be added to the study and would significantly strengthen this as a tool, or at least elaborated on in your discussion as important next steps that would advance our understanding.
Lines 259-260: Your statement about males developing body condition during reproductive periods illustrates why it is vital that you include in your methods when the BCS was assessed, please add this information and discuss how this impacts your results.

Tables and Figures:
Figure 2B – are you able to provide photographs for BCS 1 and ribs/hindquarters/tail head for BCS 2, even if not from your study? This would be very useful if someone else were to go on and use your BCS assessment criteria.
Figure 3 must have repeated observations from some of the individuals - e.g. 13 female BCS scores in year 1 when there was a maximum of three females. This needs to be taken into account in the analyses, and I also suggest using different markers per individual so that reader can see if all individuals increased in a similar way – for example, this would allow us to see if the female that did not reproduce increased in BCS faster than the two that did.
Table 1 - this may be to do with the PDF for proofing, but the layout could be improved to make text easier to read (not going on to so many lines), and could be shortened in places – for example Neck BCS 2 – you could remove the word appearance (all are based on appearance); Dewlap BCS 5 – would ‘large flap of skin’ be sufficient?
You have results repeated in Table 2 and in the text, which is typically avoided. If you keep this analysis (taking into account mu other comments), I would suggest keeping the table and only discussing the results (not presenting the statistics also) in the text. Table 2 is also missing the df for neck – these should be added for consistency.
Table 3 – as with other analyses, it would be useful to add a description of how these numbers were calculated in the methods section – how many individuals/assessments contributed to these means? Are the calves included, or just the adults?
Table 4 needs to be separated by sex (since there’s a significant difference between sexes) – actually, tables 3 and 4 could be combined if the data were reanalysed to assess changes in BCS across years, separated by sex

---

## Round 0.2 · Minor Revisions

The three reviewers have made several simple recommendations. Please revise the paper promptly and return it, when I expect I will accept it.

Reviewer 1 ·

Basic reporting

The manuscript “Body condition scoring system of reintroduced banteng (Bos javanicus D’Alton, 1923) into Salakphra Wildlife Sanctuary, Thailand” was well written and showed the reintroduction process of large herbivore as Banteng. The advantage of this manuscript is the work with reintroduction of Banteng, which is unique work. Moreover, the authors analyzed the correlation of the body condition scoring index.

Experimental design

The experimental design as good, and have added information as requested.

Validity of the findings

The findings were good with the results compared from year by year of reintroduction.
the authors added information as requested.

Additional comments

The manuscript look good now. I do not have any more comments.

·

Basic reporting

Authors have done a good job of incorporating some of our suggestions. The revised form looks much better than the original. However, the manuscript still needs a thorough proof-reading. I also recommend rewriting some of the portions. I have listed my recommendations below:
1. Some of the information mentioned in the manuscripts requires references.
2. Line 33: ...aim of the present study "was to".
3. Line 84: "the selling of this meat" sounds incorrect, kindly rephrase.
4. Line 99: The sentence is not correct. Kindly correct it.
5. Line 117: Individual's (font not correct).
6. Line 135: ‘crated’ do the authors mean created?
7. Lines 135 to 139: These sentences need to be rephrased in context to Salakphra Wildlife Sanctuary. For example, the sentence “The height above sea level is between 700 and 1,000 m.” is incomplete, the height of what? This can be rephrased as: “The altitude of the study site ranges from 700 to 1,000 m asl.”
8. Lines 161 to 162: It is not clear what authors wanted to convey. This sentence should be modified.
9. Lines 181 to 182 end without ‘full-stop’.
10. Lines 182 to 183: The sentence is not correct. Kindly re-write it.
11. Line 224: It is not clear who was the inter-assessor? What does ‘variate’ mean in this sentence? This sentence needs re-phrasing.
12. Line 235: “seasons”
13. Lines 288 to 289: “seasons”
14. Lines 297-298: there are multiple ‘key’
15. Most of the figure legends are incomplete. Kindly correct them.

Experimental design

1. Lines 32 and 33: The sentence is too abrupt. Authors are instantly talking about the influence of body condition, without mentioning why reintroduction is needed for the banteng.
2. The last paragraph of "Introduction" still does not do justice in mentioning the hypothesis and objectives clearly. Kindly re-write.
3. Statistical analyses: I appreciated the authors for including this section. It would be good if authors could mention what kinds of data (categorical/ordinal/continuous) were collected and why those statistical tests were used.

Validity of the findings

Authors have now incorporated some of the suggestions on validating the BCS through camera-traps and using the inter-assessor differences. The results now look promising. However, what concerned me was the discussion. The discussion should properly highlight the importance of findings and what it means in the context of other studies. There are lots of grammatical errors in this section and also the sentence formations are incorrect. I have listed some of my concerns pertaining to the Discussion section below:
1. Lines 277 to 279: these sentences are not correct.
2. The first sentence of the discussion should highlight the findings or what the study indicated.
3. Lines 289-290: “This finding indicated high resource quality of the area as recommended by Pokharel et al. (2017)” this sentence does not reflect why it was used. Do the authors mean the “influence” on BCS? Kindly rephrase it.
4. Lines 290-293: This sentence is long and not correct. It needs complete rephrasing.
5. Line 323: Authors can include one of the publications on Asian elephants highlighting the influence of lactation on the body condition (Pokharel, S. S., Seshagiri, P. B., & Sukumar, R. (2020). Influence of the number of calves and lactating adult females in a herd on the adrenocortical activity of free-ranging Asian elephants. Wildlife Research, 46(8), 679-689.)
6. Lines 324-325: why authors highlight that the ‘fluctuations’ in BCS is due to mortality? Authors are recommended to provide references here.

Additional comments

The revised version looks better than the original manuscript. But it appears that the authors did not proof-read the discussion section. So kindly proof-read the manuscript and rephrase the sentences. The results are promising. But these results should be well-highlighted in the manuscript.

Reviewer 3 ·

Basic reporting

There are still some spelling and grammatical errors that should be corrected - in particular, please spell check the manuscript for banteng (in a few places is spelled bateng), check for BSC instead of BCS. Additionally, I feel the discussion could benefit from a second review for sentence structure.

Lines 136-139: please review this sentence – remove second ‘of’ after introduced
Line 141, 142, 275: please check the spelling of banteng (here is bateng) throughout
Line 539: Change BSC to BCS – please check for this error throughout
Still some editing for English language required, particularly the discussion

Experimental design

Please explain the inter-observer reliability assessment in more detail

Lines 490-493: Suggest: 'Inter-observer reliability was assessed using BCS from the training cage and the captured photographs, which were reanalyzed by the observer and other assessors' and add details of how this was assessed, what were criteria for passing - was there a percentage of images that all observers had to agree on before proceeding?

I am still concerned that there is no explanation of how many images per individual were used in each year, and no notable consideration of how repeated observations on a few individuals may have impacted the results. The authors have provided the total number of BCS scores and total number of individuals, but not how many BCS scores per individual per year. As per my previous review, if some individuals are over-represented in the data, this could bias your results i.e. pseudoreplication. At the very least, this should be added to the discussion as a potential limitation, not ignored. It is clear from Figure 4 that individual variability in BCS was potentially quite high - the third year only had one female represented, and all data points are fairly tightly distributed, whereas year 2 had two females and a much wider range. This variability makes it difficult to see whether all three females did indeed increase BCS across the study period, or perhaps only the female that did not breed gained condition. It would be possible to use different markers for each individual (and open or closed for male and female) then we can at least visually determine whether all four females and three males increased BCS over time.

One benefit of conducting BCS via photographs is that you can identify individual animals and peotentially investigate why some might be adapting better than others to the habitat - i.e. dominant males might do better than subordinate if the habitat is sub-optimal, breeding females may lost more condition than non-breeding females etc. So I think the ability to look at individual differences is not a bad thing here.

Also please note that Figure 4 legend states red dots for both males and females, which should be corrected.

Figure 3 legend - please add the samples numbers used in this figure (both number of individuals and BCS scores per individual)
Top section of figure 2 - should say ribs, not rips

Validity of the findings

Please add more detail regarding the criteria if inter-observer assessment, and discuss that the small sample size and individual differences in BCS change over the study period may influence your results. This does not take away from the usefulness of the method or the study, but makes it easier for the reader to interpret.

Additional comments

Thank you for responding to my previous comments - there are still a few of my concerns that remain, so please consider including these in your final manuscript.

---

## Round 0.3 · Minor Revisions

I don't have any problems with the content of this, I think you have answered all the reviewers' concerns. However, the writing does need substantial improvement — as noted by PeerJ's editorial staff. You should either work with them to get help or find a native English speaker who can help you. There are computer packages — not just Word and its grammar checker, but Grammarly, both of which I use extensively that might help. The revision does have to meet PeerJ's editorial standards.

---

## Round 0.4 · accepted · Accept

Thank for you correcting the English.